# Customer Value Types Predicting Consumer Behavior at Dutch Grocery Retailers

**DOI:** 10.3390/bs10080127

**Published:** 2020-08-05

**Authors:** Kim Janssens, Wim Lambrechts, Henriëtte Keur, Janjaap Semeijn

**Affiliations:** Faculty of Management, Department of Marketing and Supply Chain Management, Open Universiteit, Valkenburgerweg 177, 6401 DL Heerlen, The Netherlands; wim.lambrechts@ou.nl (W.L.); henriette.keur@outlook.com (H.K.); janjaap.semeijn@ou.nl (J.S.)

**Keywords:** consumer behavior, customer value, grocery retail formats, satisfaction, repurchase intention, Word-of-Mouth

## Abstract

The purpose of this paper is to profile three grocery retail formats (non-discounter, soft discounter, and hard discounter) in the Netherlands using Holbrook’s value types. These value types are linked to three consumer behavior outcomes, i.e., Satisfaction, Repurchase intention, and Word-of-Mouth. The impact of the retail formats is evaluated on the importance and performance of the value types, using a questionnaire for each of the retail formats. The relationship between the value types and outcomes is tested with partial least squares structural equation modeling (PLS-SEM). Aesthetics, Altruistic value, and Efficiency are positively linked to Satisfaction. In addition, Efficiency is the key driver for Repurchase intention and has a positive impact on Word-of-Mouth. A positive Word-of-Mouth is predicted by Social value and Service excellence. The three examined retail formats show a difference in Holbrook’s value types. Overall, the results of the non- and soft discounters differ from the hard discounters. Remarkably, and contrary to previous studies, the soft discounter’s scores are the highest meaning that consumers are most critical for this retail format. It seems that consumers expect the best of both worlds at soft discounters: low prices, interesting bargains, easy access, but also appealing store design, and excellent service.

## 1. Introduction

Nowadays, consumers not only have a wide variety of choice in food products, but also a broad range of grocery stores to choose from. In the Netherlands, there are about 6338 grocery stores and this number is still growing [1]. In 2019, the turnover of Dutch grocery stores exceeded 40 million euros. Whereas prices in specialty stores are relatively high, grocery stores offer a myriad of brands and generics making shopping for food affordable to all [2]. However, all these options make grocery shopping a complex experience and may cause shopper anxiety due to the overwhelming choice. Having to make choices about what, when, and where to shop makes consumers feel frustrated and uncertain [3]. Many aspects play a part in making this decision, but designing experiential marketing strategies to create greater customer value through consumption has definitely become one of the aspects by which retailers can differentiate themselves [4]. As a key concept in experiential marketing and services research, customer value plays a fundamental role in customers’ decision-making and evaluation processes [5]. Besides being crucial in the pre-purchase process of deciding customer value has been studied regarding its impact on post-purchase judgments like satisfaction, repurchase intent, and word-of-mouth [6,7]. Hence, meeting consumers’ needs and wants and creating customer value can be used as a competitive advantage in influencing consumers’ buying behavior.

Especially in grocery retailing, competition is fierce with discounters making it hard for non-discounters by gaining ground and establishing a large market share [8]. Consumers constantly make a trade-off between the benefits and costs of a particular (grocery) retailer in order to decide where to buy. The current research is a follow up/extension of the research of Willems et al. [7] examining customers’ perception of value in three different retail formats. As these authors already stated, the application of customer value in grocery retailing is rather limited. Previous studies have examined customer value in hospitality services [9], from a product-oriented approach [7], as an interaction between customers and their experience with a parcel locker [10], in restaurant settings [11], mobile services [12], fashion retail [13], but also in customer participation [14], co-production [15] and gamification [16]. In order to elaborate and generalize findings on how customer value differs in retail formats, this study responds to recommendations of Willems et al. [7] to conduct a similar study in other countries. While their research took place in Belgium this study was set up in the Netherlands, a country with a high supermarket density.

The purpose of this research paper is (1) to analyze three different Dutch retail formats in terms of seven value types (Efficiency, Product and Service excellence, Social value, Play, Aesthetics, and Altruistic value) based on Holbrook’s value typology; (2) to link these value types to consumer behavior outcomes (i.e., Satisfaction, Repurchase intention, and Word-of-Mouth); (3) to evaluate the impact of the retail formats on performance and importance of the seven value types.

This paper is structured as follows. After this introduction, Section 2 gives an overview of relevant literature on customer value alongside a description of Dutch grocery retailer formats. In Section 3, the methodology is described, followed by Section 4 presenting the results of the study. Section 5 and Section 6 conclude with the main findings and discussion on implications of the results, limitations of the study, and suggestions for future research.

## 2. Literature Review

### 2.1. Customer Value from an Experiential Marketing Perspective

According to Batat [17], customer value can be seen as a result of customer experiences largely depending on the personal perception of consumers. Hence, customer value can be economically, functionally, individually, and socially defined with regard to certain customer experiences. Although it is a broadly used concept in marketing studies, a clear cut definition has not yet been established and therefore, cornering customer value is still ground for discussion [18]. When browsing through marketing literature many classifications can be considered [19,20]. In the current study, customer value is described as value-in-experience [17] following Holbrook’s typology of experiential customer value, which he conceptualizes as [21] (p. 5) “an interactive relativistic preference experience”. Holbrook’s framework relies on three underlying dimensions: (1) extrinsic vs. intrinsic value (value perceived as using a good or service as a means to an end vs. as using it as an end-in-itself), (2) self- vs. other-oriented value (value perceived as for the consumer’s own benefit vs. as for an other’s benefit) and (3) active vs. reactive value (value perceived as through the direct use of a good or service vs. as the response on the good or service). Using these three dimensions, Holbrook developed a matrix representing eight types of customer value: Efficiency, Excellence, Status, Esteem, Play, Aesthetic Value, Spirituality, and Ethics. As Status and Esteem are interrelated and difficult to operationalize separately, later studies combined the two in an overarching category labeled Social value [22,23,24,25]. In a similar vein, Ethics and Spirituality are combined in Altruistic value [25]. Bourdeau et al. [23] and Du et al. [25] define Altruistic value as customers’ beliefs that—in a retail context—the store is socially responsible. In addition, Holbrook’s Excellence is split into two separate constructs: Product excellence and Service excellence [23], given retailers offer a mix of products and services [26].

Customer value is an important antecedent of consumer behaviors like Satisfaction, Repurchase intention, and Word-of-Mouth [6,27,28]. Satisfaction is an objective for managers of retail establishments because high levels of satisfaction lead to the establishment of profitable relationships over time [29,30]. Satisfaction could be described as an effective response of varying intensity (specific/accumulative satisfaction) [31,32]. Research by Nilsson et al. [33] showed that satisfaction is higher for fill-in grocery shopping (i.e., when just a few items are needed) than major grocery shopping (stocking up), that time pressure has no effect on satisfaction, and that price level, service quality, and product quality/range are more important for satisfaction with major grocery shopping, whereas access is more important for satisfaction with fill-in shopping. It is also found that the importance of attributes for satisfaction depends on the type of shopping more than on individual characteristics. Customer value Efficiency proved to be the key predictor of customer satisfaction, which is in line with the task-oriented nature of grocery shopping [34,35]. In agreement with Willems et al. [7], behavioral intention is separated in Repurchase intention and Word-of-Mouth. Repurchase intention is defined as the willingness of the shopper to use and purchase in the supermarket again [7]. Loyalty behavior of consumers is also presented by repurchase intention according to Fuentes-Blasco et al. [36]. A definition of Word-of-Mouth according to Westbrook [37] (p. 261) is the following: “all informal communications directed at other consumers about the ownership, usage, or characteristics of particular goods and services or their seller”.

In the literature, several influences of Play on consumer behavior are reported. Play lowers customers’ satisfaction, but positively impacts Word-of-Mouth and Repurchase intention [36]. The negative effect of Play on Satisfaction is in contrast with previous studies [38,39,40] indicating a positive, significant relationship between perceived hedonism and customer satisfaction. Conversely, the positive effect of Play on Repurchase intention and Word-of-Mouth implies that shoppers do appreciate a recreational aspect when choosing a particular store in the future, and when talking to other people [36]. Store-based attributes in the food retail industry have a different influence on customer satisfaction and loyalty depending on the quality perception of products, with a moderating role of products’ perceived quality [41]. Another aspect with positive effects on customer loyalty is consumers’ perceptions of retail business ethics such as social responsibility. Furthermore, customers’ perceptions of retail business ethics exert a stronger effect on consumer trust in integrated social discount spaces, though social discount practices do not affect the link between such perceptions and loyalty [41]. Compared to when the retailer does not offer discount space, collaborative and integrated social discount spaces have weaker effects on trust and loyalty [42]. Although the above-mentioned research touches upon the influence of several customer value types on Satisfaction, Repurchase intention, and/or Word-of-Mouth, literature does not yield consistent results. Therefore, the current study builds upon Willems et al. [7] and examines the impact of Holbrook’s seven customer value types on behavioral outcomes Satisfaction, Repurchase intention, and Word-of-Mouth.

From this perspective, this study will test the following hypotheses:

**H1a.** 
*Aesthetics positively influences Satisfaction*


**H1b.** 
*Aesthetics positively influences Repurchase intention*


**H1c.** 
*Aesthetics positively influences Word-of-Mouth*




**H2a.** 
*Altruistic value positively influences Satisfaction*


**H2b.** 
*Altruistic value positively influences Repurchase intention*


**H2c.** 
*Altruistic value positively influences Word-of-Mouth*




**H3a.** 
*Efficiency positively influences Satisfaction*


**H3b.** 
*Efficiency positively influences Repurchase intention*


**H3c.** 
*Efficiency positively influences Word-of-Mouth*




**H4a.** 
*Play positively influences Satisfaction*


**H4b.** 
*Play positively influences Repurchase intention*


**H4c.** 
*Play positively influences Word-of-Mouth*




**H5a.** 
*Product excellence positively influences Satisfaction*


**H5b.** 
*Product excellence positively influences Repurchase intention*


**H5c.** 
*Product excellence positively influences Word-of-Mouth*




**H6a.** 
*Service excellence positively influences Satisfaction*


**H6b.** 
*Service excellence positively influences Repurchase intention*


**H6c.** 
*Service excellence positively influences Word-of-Mouth*




**H7a.** 
*Social value positively influences Satisfaction*


**H7b.** 
*Social value positively influences Repurchase intention*


**H7c.** 
*Social value positively influences Word-of-Mouth*


### 2.2. Dutch Grocery Retailer Formats

Although the grocery landscape is rapidly changing globally [43], the Dutch grocery retail market remains quite solid. The two largest food retailers in the Netherlands control about 55% of the market and are non-discounter supermarkets, Albert Heijn and Jumbo. The non-discounter retail channel in The Netherlands is dominated by national and local players contrary to the discounter channels [44]. Non-discounters have the benefit of offering a wide range of food and non-food products, premium brands, and private labels. However, given the modern consumers’ cautious spending pattern they often choose to shop for groceries at different supermarkets often advantaging discounters. In many European countries, discounters hold considerable market share and are the fastest-growing retail format. In The Netherlands, discounters account for 65% of food retail (with 19% soft discount and 46% hard discount) [45]. Discounters Aldi and Lidl (Germany based) have the strongest growth in the Netherlands, being the main operating discounters in the market [7].

In order to gain competitive advantages, retail formats make adjustments and adapt to fit the needs of the consumer. Discounters typically offer a higher relative proportion of private labels, smaller shopping areas, limited ranges of food products, a smaller assortment of non-food products to sell, and lower price compared to non-discounters [46,47]. To offer lower prices, they use a simplified “no-frills” store format, a “what you see is what you get” approach [47]. Within the category of discount retail services hard and soft discounters are distinguished [7]. Hard discounters offer a limited assortment consisting mainly of own store brands (private labels), have limited in-store fixtures, and offer limited services [48,49]. Conversely, soft discounters have nearly all product categories in their assortment, offer both private labels and national brands, and offer a wider range of fresh, chilled, and frozen food than hard discounters [49,50]. In addition, over the past years, soft discounters have changed their strategy and offer a limited range of fresh food products and premium brands at lower prices than at non-discounters [7].

Willems et al. [7] stated that shoppers’ perceptions are different in terms of the customer value that each of these retail formats puts forward. Each retail format can (and should) correspond to a different positioning in customers’ minds. By using Holbrook’s comprehensive value typology, Willems et al. [7] examined whether different retail formats are indeed perceived differently in the minds of customers. The results of the research indicate that retail formats indeed significantly differ in terms of customer value types. Overall, the hard discounter obtains the lowest average scores and is significantly different in the customer’s perception regarding all seven customer value types, compared to both the non-discounter and the soft discounter. The differences between the soft discounter and the non-discounter are far less pronounced. The only statistically significant differences are for Play and Aesthetics, for which the non-discounter outperforms the soft discounter. While Willems et al. [7] expected discounters to perform better on price, they perform worse in terms of opening hours, one-stop shopping opportunities, and hassle-free shopping carts, and thus in terms of overall Efficiency. There are, however, two statistically significant cross-format differences identified regarding the structural relationships between the hard and soft discounter: (1) between Efficiency and Satisfaction and (2) the relationship between Social value and Word-of-Mouth intentions. In both cases, the relationship is significantly stronger for the hard discounter. Willems et al. [7] examined how these three retail formats put different customer value types forward in Belgium. As discounters and non-discounters are, too, the main grocery purchasing channels in the Netherlands, we follow previous research and examine which customer value types can be associated with the three retail formats under study. Leaning on previous findings, we hypothesize that depending on the retail format, customers have different expectations and therefore prioritize customer value types accordingly:

**H8.** *The relationship between the customer value types (Aesthetics, Altruistic value, Efficiency, Play, Product excellence, Service excellence, Social value) and consumer outcomes (Satisfaction, Repurchase intention, Word-of-Mouth) is moderated by retail format type; the relationship is stronger for non-discounters and soft discounters than for hard discounters*.

Willems et al. [7] were the first to disentangle customer value in a grocery shopping context and the current paper is based upon one of their recommendations (pp. 606–607): “To generalize the results (…) to conduct similar studies in other countries as well, as there may be cultural differences with regard to the strategic importance of the different value types”. Hence, this research extends their findings and analyzes three grocery retailer formats (hard, soft, and non-discount retailers) in the Netherlands. Although Belgium (context of [7]) and the Netherlands (the focus of this study) are neighboring countries and share a common language, [51] discovered large differences in customer orientations between Belgium and the Netherlands. This (mainly cultural) difference may perhaps also lead to differences in evaluating customer value types in the specific retail context of grocery shopping. In this study, we selected non-discounter Albert Heijn, soft discounter Plus, and hard discounter Lidl, all located in the city of Amersfoort in the Netherlands. Figure 1 illustrates the conceptual model.

## 3. Materials and Methods

### 3.1. Procedure

The survey taker handed out the questionnaire to adult consumers (age > 18 years old) at a primary school, three sports clubs, and people in the survey taker’s network in the spring of 2018 in Amersfoort, a middle-sized city of 150,000 inhabitants in The Netherlands. This sample was a non-probabilistic convenience sample.

Six persons were asked to pre-test the survey to identify ambiguities and to critically evaluate the questions on content and wording. Based on this feedback, some questions were re-formulated or slightly modified in order to be more precise and clear.

### 3.2. Measures

Customer value was measured with the value typology construed by Willems et al. [7] (see Table 1), extending work done by Holbrook [21].

Respondents were asked to indicate on a 5-point Likert scale (from strongly disagree (1) to strongly agree (5)) to which extent they agreed with the 90 statements in the questionnaire. For an overview of all the statements, see Appendix A.

To conduct the research as closely as possible to work performed in Belgium [7], for this study too the market leaders of the various grocery retail formats in the Netherlands were selected. That is, respondents needed to rate one of the following grocery retailers: non-discounter Albert Heijn, soft discounter Plus, or hard discounter Lidl.

Seven customer value types were defined, with either having reflective or formative measurement models [60]. Altruistic value, Play, Social value (reflective) and Aesthetics, Efficiency, Product excellence, Service excellence (formative). Partial least squares structural equation modeling (PLS-SEM) was used to test the relationship between each of the seven types and the three key outcomes. The conceptual model which is central to this study is rather complex given the large number of indicators and relationships. PLS-SEM can handle more complex models [61] and allows analysis using a smaller sample [62].

## 4. Results

### 4.1. Participants

A total of 147 questionnaires was handed out. The final sample consists of 126 respondents (n = 62 for the non-discounter, n = 31 for the soft discounter and n = 33 for the hard discounter). Age ranged from 20 to 70+, with the majority being between 30–40 years old and female (69%), see Table 2.

### 4.2. Evaluation of Measurement Models

For readability purposes, we will start by discussing the results of the reflective measurement models, followed by those of the formative measurement models.

#### 4.2.1. Reflective Measurement Models

A first step in evaluating the reflective latent variables (i.e., Altruistic Value, Play, and Social Value) is measuring internal consistency by means of Cronbach’s alpha and composite reliability (CR). Results are well above the threshold of 0.70, indicating sufficient internal consistency reliability (Table 3).

The next step is measuring convergent validity, i.e., the extent to which indicators of the same variable share a (high) proportion of variance. To estimate this relationship between the latent variables (i.e., Altruistic value, Play, and Social value) and their indicators the outer loadings of the indicators are calculated. The higher the outer loadings on a construct or variable, the more the associated indicators have in common. All loadings are well-above the threshold of 0.70, suggesting a sufficient level of indicator reliability [63].

To evaluate convergent validity (i.e., a positive correlation of an indicator with another indicator of the same construct) of reflective constructs the outer loadings of the indicators and the average variance extracted (AVE) are considered. The outer loadings are 0.69 or higher—sufficiently close to the threshold of 0.70 as to not to be removed since deleting the indicators <0.70 does not lead to an increase in composite reliability. All AVE measures are above 0.50 (Table 3) indicating that the convergent validity of each latent variable is acceptable.

In the last step of evaluating the reflective measurement models, discriminant validity is determined. A construct needs to be adequately distinct from other constructs in the model. Three measures are applied to analyze a construct’s discriminant validity. Assessing the construct’s indicators’ cross-loadings is the first approach [63]. In order to be sufficiently distinctive, an indicator’s outer loading on the linked construct needs to be greater than any of its cross-loadings on the other constructs. Table 4 shows that an indicator’s highest loading is expected with the construct it has been assigned to.

The Fornell-Larcker criterium is the second approach to check for discriminant validity (see Table 5). The average variance shared between a construct and its measures should be greater than the variance shared between the construct and the other constructs of the model. Discriminant validity is considered adequate when the square root of the average variance extracted (AVE) for a construct is higher than the correlations between that construct and the other constructs [63]. The square root of the AVE of each construct is to be read on the diagonal and the off-diagonal holds the correlations between the constructs. Table 5 shows that the square root of the constructs’ AVE is higher than the correlations between constructs.

Thirdly, the heterotrait-monotrait ratio (HTMT) of the correlations is assessed [64], i.e., the ratio of the between-trait correlations to the within trait correlations. A threshold of 0.85 is kept to ensure adequate distinct constructs. In this model, all HTMT results are below 0.55 and thus acceptable. The bootstrapping procedure corroborates these findings, showing confidence intervals not containing the value 1, suggesting that the constructs are empirically distinct.

#### 4.2.2. Formative Measurement Models

First, convergent validity needs to be determined for Aesthetics, Efficiency, Product excellence, and Service excellence by carrying out separate redundancy analyses for each construct [63]. These analyses yield path coefficients of 0.94 and higher, well above the recommended threshold of 0.70 thus supporting the formative constructs’ convergent validity.

Next, collinearity between indicators is assessed by looking at the outer VIF values. The results show that the indicator values for Product excellence, Efficiency, and Aesthetics are well below the threshold of 5. Service excellence, however, yields a value of 6.7 for indicator serv_9. Since the remaining indicators sufficiently capture the construct’s content, serv_9 is removed from the construct (and model). Next, the outer weights are considered for relevance (resp. relative and absolute importance) and the indicators’ significance is assessed via bootstrapping. When the formative indicators’ outer weights are not significant removing these indicators is strongly advised [65]. Therefore, six formative indicators were removed from analyses (i.e., aesth_2, eff_16, effi_17, prod.exc_5, prod.exc_6 and serv_9).

After eliminating six indicators in total, the remaining reflective and formative constructs have satisfactory levels of quality to proceed to the evaluation of the structural model.

### 4.3. Evaluation of the Structural Model

To check for collinearity the indicators’ Inner variance inflation factor (VIF) values are considered [63]. All VIF values are below the threshold of 5 and hence, collinearity is not an issue. Next, **R^2^** measures the predictive accuracy of the structural model. This coefficient of determination indicates the variation in Repurchase intention, Satisfaction, and (positive) Word-of-Mouth, explained by the independent variables. A higher R^2^ means more variability is explained by the structural model. The effect ranges from 0–1, with ‘1′ indicating full predictive accuracy. In this study, R^2^ has values of respectively 0.35, 0.46, and 0.47; in social science effect sizes as of 0.04 are considered the recommended minimum representing a practically significant effect [66]. Effect sizes between 0.30 and 0.50 can be seen as weak to moderate. This indicates that all three outcome variables are moderately predictive. In addition, effect size (f^2^) is determined by the change in R^2^ when a construct is eliminated from the structural model [63]. Values of 0.02, 0.15, and 0.35 respectively indicate small, medium, and large effects while values < 0.02 indicate there is no effect. Table 6 shows the relevant effect sizes.

Path coefficients show the structural model’s relationships among the constructs (see Table 7). Looking at the relative importance of the driver constructs for the consumer behavior outcomes (i.e., Repurchase intention, Satisfaction, and Word-of-Mouth), Efficiency appears to be the most important driver for customers’ Repurchase intention (0.42) and Satisfaction (0.31). Service excellence is the main driver for a positive Word-of-Mouth (0.24). Apparently, the only construct that is negatively related to both Repurchase intention and Satisfaction is Social value.

When taking the outer weights into consideration, the most important indicators within the driving constructs can be addressed. That is, when taking a closer look at Efficiency three indicators are mainly responsible for the effect, i.e., indicators concerning price-quality ratio and indicators on store layout (accessibility). Service excellence is mainly influenced by indicators on personnel approachability (for an overview of all constructs and their indicators see Appendix A).

To assess whether the above-mentioned relationships are significant a bootstrapping procedure is run. Table 8 shows the significant relationships in bold. Figure 2 presents the final model.

Aesthetics has a positive, significant effect on Satisfaction, which is in line with H1a. Similarly, Altruistic value has a positive, significant effect on Satisfaction, supporting H2a. Table 8 shows that Efficiency is a positive, significant predictor for all three consumer outcomes (Satisfaction, Repurchase intention, and Word-of-Mouth) which confirms H3a, H3b, and H3c. Service excellence, as well as Social value, have a positive, significant effect on Word-of-Mouth, backing up respectively H6c and H7c. No significant differences were found for the moderating effect of retail format type on the relationship between customer value type and consumer behavior outcome. Table 9 presents an overview of all hypotheses results.

### 4.4. Relating Customer Value Perceptions in Different Retail Formats to Customer Behavior Outcomes

To access the relative performance of the different retail formats on each customer value type, the path coefficients of the model are compared in the overview of Table 10.

Although we did not find moderating effects, from Table 10 can be concluded that the three retail formats differ in path coefficients for value perceptions Aesthetics, Altruistic value, Efficiency, Service excellence, and Social value on the outcomes Satisfaction, Repurchase intention and Word-of-Mouth.

Aesthetics has an overall positive, significant effect on Satisfaction; yet, to keep customers satisfied it appears to be more important for soft discounters and non-discounters to have a pleasant looking retail environment than it is for hard discounters. Given the negative sign, customers seem to be more satisfied with a non-aesthetically pleasing hard discounter. Altruistic value has an overall positive, significant effect on Satisfaction. That is, customers indicated to be more satisfied when the hard discounter engaged in socially responsible retailing than did the other two retail formats. Efficiency has an overall, positive significant effect on all three consumer behavior outcomes. The path coefficients of Efficiency are the largest for the soft discounter regarding Satisfaction and Repurchase intention. Efficiency also seems to be more important for Repurchase intention than for the other two behavior outcomes. Service excellence has an overall positive, significant effect on Word-of-Mouth. It seems that at soft discounters customers expect excellent service from the personnel in order to create a positive Word-of-Mouth, more than they expect this at the other two retail formats. Social value has an overall, positive significant effect on Word-of-Mouth and results show that this value type is equally more important for both discounters than for the non-discounter.

Both Play and Product excellence did not yield any effect on customers’ Satisfaction, Repurchase intention, or (positive) Word-of-Mouth, for none of the three retail formats.

### 4.5. Additional Analyses: Customer Value Perceptions in Different Retail Formats

Given the limited sample size and the restriction in SmartPLS of not providing an Omnibus test of group differences (OTG) yet, additional analyses were performed in SPSS. To test differences in customer value perceptions between the three retail formats a series of Anovas was performed. A between-subjects test revealed a significant difference in perception of Product excellence (*p* < 0.05). More specifically, pairwise comparison yielded a significant difference between the soft and hard discounter. The soft discounter scores significantly higher (M = 4.02, SD = 0.08) on Product excellence than the hard discounter (M = 3.17, SD = 0.08), F(2, 123) = 32.75, *p* < 0.001. In addition, significant differences were found between all retail formats for Aesthetics, F(2, 123) = 6.54, *p* < 0.01. Both the non-discounter (M = 3.75, SD = 0.06) and soft discounter (M = 3.82, SD = 0.09) scored significantly higher on Aesthetics than the hard discounter (M = 3.42, SD = 0.08). The non-discounter scored slightly lower on Aesthetics than the soft discounter, but not significantly. Altruistic value, Efficiency, Play, Service excellence, and Social value did not differ across the three retail formats.

## 5. Discussion

Aesthetics has an overall positive, significant effect on customers’ Satisfaction. This result differs from Willems et al. [7], but is in agreement with the findings by Terblanche [67] and Fuentes-Blasco et al. [36]. That is, Willems et al. [7] stated that Aesthetics influences loyalty behavior, both in terms of Repurchase intention and Word-of-Mouth but has no impact on Satisfaction. Conversely, Terblanche [67] and Fuentes-Blasco et al. [36] found that the factor “internal shop environment” of the in-store shopping experience for supermarket customers has a positive, significant relationship with customer satisfaction. Although the overall results indicate a positive influence, the scores do differ, although not significantly, between the retail formats. Clearly, it is more important for non-discounters and soft discounters to have an appealing store design than it is for hard discounters to keep customers satisfied. Even more, a negative relationship between Aesthetics and Satisfaction was found for the hard discounter. Apparently, consumers expect a hard discounter to be straightforward, without any frills.

The overall positive, significant influence of Altruistic value on Satisfaction is in alignment with other aspects of customers’ perceptions of retail business ethics such as social responsibility, having positive effects on customer loyalty [41]. The current study thereby corroborates findings of IRI [45] stating that over 60% of Dutch shoppers identify themselves with fairness, respect for the environment, and packaging. However, a (non-significant) difference was found between the retail formats. The soft discounter scored negatively on Altruistic value for the outcome Satisfaction. Being socially responsible is perceived as negative for customers’ Satisfaction in soft discounters, contrary to the hard discounter and the non-discounter showing a positive connection.

Whereas Wagner et al. [34], Esbjerg et al. [35], and Willems et al. [7] concluded that Efficiency was a key predictor for Satisfaction, the current study showed that Efficiency is the key predictor for Repurchase intention and a significant predictor for Satisfaction, in agreement with Fuentes-Blasco et al. [36]. In addition, Efficiency also has an overall positive, significant impact on Word-of-Mouth. Working efficiently, having a clear store layout, and easy product access are significantly more important for the soft discounter to engage customers in repurchasing and keeping them satisfied than for the non-discounter and hard discounter.

Service excellence is a key predictor for Word-of-Mouth. Willems et al. [7] also found a positive significant effect of Service excellence on Word-of-Mouth. Especially, indicators asking about personnel approachability have a high score. It seems that having friendly and helpful personnel is even more important for soft discounters: to create a positive Word-of-Mouth they need to offer an excellent service. Perhaps, with a non-discounter customers automatically assume better service whereas they do not expect extra service with a hard discounter. Hence, soft discounters fall in-between and customers seem to demand the best of both worlds: competitive pricing and quality, and decent service.

Social value is negatively connected to Repurchase intention and Satisfaction and has a positive, significant impact on Word-of-Mouth. Contrary, Willems et al. [7] found a positive, significant impact on Repurchase intention and Satisfaction. It suggests that the symbolic value of the store patronage decision remains an issue, even in a grocery shopping context. This positive result is related to previous studies focusing on the strategic role of store (format) personality and self-congruity [68,69,70,71]. Store personality is how a store is captured in a consumers’ mind and consists both of the store’s functional attributes and psychosocial elements [72]. Social value, therefore, seems to be more important to create a positive Word-of-Mouth for both discounters than for the non-discounter. It appears that especially for discounters efforts made towards responsible retailing are highly appreciated. The minor effect of Social value on the customer key outcomes is in agreement with the expectation regarding the importance of Social value in countries where (hard) discounters are more and more common [7].

In the literature, several influences of Play and Product excellence on customers’ Satisfaction, Loyalty (e.g., repurchase behavior), and Word-of-Mouth are reported. However, Play and Product excellence show a rather limited role in affecting the three consumer behavior outcomes in the current study.

With the results in mind, there are a number of ways in which these may have practical implications. First, customer value Aesthetics is important for non-discounter and soft discounter, especially when it comes to keeping customers satisfied. A grocery store, therefore, should pay attention to following in-store aspects improving overall Aesthetics:Appropriate lighting in the store, i.e., matching the light settings to the specific function of a certain category. For instance, fruits and vegetables benefit from bright illumination to create a fresh outdoor market feeling whereas the bakery corner demands more luxurious lighting [73].Pleasant and appropriate appearance of the staff. Customers also need to be able to distinguish employees from other customers to have a positive customer experience.An appealing supermarket layout (e.g., equipment, design, decoration, furniture).

Second, Altruistic value seems somewhat more important for hard and non-discount retailers than for soft discounters. The former can increase Satisfaction with their consumers when explicitly highlighting the store’s social conscience. Putting socially responsible actions in the spotlight and communicating the store’s efforts towards citizenship enhances customers’ satisfaction and will add to a positive customer experience.

Third, for all retail formats and particularly for soft discounters it is important to pay attention to Efficiency. Efficiency is positively influencing all consumer behavior outcomes and retailers can improve this in-store by:Offering products with a fair price-quality ratio.Having a store layout that makes it easy to find the products customers are looking for.Designing the aisles in the store in such a way customers can move smoothly through the store.

Fourth, Service excellence is more appreciated in soft discounters than in the other two retail formats under study and especially affects Word-of-Mouth. Retailers, therefore, should pay extra attention to briefing and educating their staff with regard to approachability. Customers engage in a positive Word-of-Mouth when personnel is helpful when they solve complaints and resolve potential problems. When training their staff, retailers must focus on these particular aspects to create an excellent customer service.

Fifth, a positive Word-of-Mouth can be established for discounters when addressing the social value of the store. Although shopper perceptions of discounters are changing, discount retailers often still struggle with the image of offering lower quality or attracting a certain target group [74]. Nothing is less true, but retailers must be aware of these perceptions and should emphasize specific strengths in their marketing communication campaigns.

This research has some limitations. The first limitation is the number of respondents that filled out the questionnaires. The estimated path coefficients between the customer value types and the consumer behavior outcomes could be improved by increasing the sample size. In addition, to verify whether the relationship between the customer value types and the consumer outcomes varies across the retail formats, a multigroup analysis should be performed. Given the limited sample size, this was not possible with the current dataset.

Second, especially now retailers (including grocery stores) increasingly engage and invest in using in-store consumer experience to create competitive advantage, Play and how retailers give form to this construct seems worthwhile examining in future studies. The current study focused on the experience in-store and related customer value types as encountered in the brick-and-mortar shop but it would be interesting to test whether other perceived values are important when shopping for groceries online.

Third, Service (excellence) was now measured with statements assuming an interaction between customer and personnel but it may as well be an interaction between a customer and (in-store) technology. It is plausible that the elements that now make service excellence are quite different when robots are filling the aisles or being deployed as cashiers in supermarkets.

Besides the role of different retail formats having an influence on the relationship between customer value types and consumer behavior outcomes, other moderators could be examined. For instance, it would definitely provide more insights into shoppers’ profiles when adding the type of shopping (e.g., routine shopping or fill-in shopping trip) to the model.

## 6. Conclusions

This study gives insights into customers’ value perception for three different Dutch grocery retail formats. The purpose of this study was (1) to evaluate three Dutch grocery retailers in terms of Holbrook’s seven value types (Aesthetics, Altruistic value, Efficiency, Play, Product excellence, Service excellence, and Social value); (2) to link these value types to three consumer behavior outcomes (i.e., Satisfaction, Repurchase intention, and Word-of-mouth); (3) to evaluate the impact of the retail format on performance and importance of the seven value types.

Regarding the relative importance of some of the value types in the driving key outcomes for Dutch grocery retailers, the strongest, significant relationships are summarized:Satisfaction is positively influenced by Aesthetics (β = 0.22, t = 2.13), Altruistic value (β = 0.23, t = 2.29), and Efficiency (β = 0.31, t = 3.11), all *p* > 0.01.Repurchase intention is positively influenced by Efficiency (β = 0.49, t = 4.29), *p* < 0.001Word-of-mouth is positively influenced by Efficiency (β = 0.22, t = 2.01), Service excellence (β = 0.29, t = 2.26), and Social value (β = 0,19, t = 1.95), both *p* < 0.05.

The bootstrapped R^2^ (R^2^_Satisfaction_ = 0.46, R^2^_Repurchase intention_ = 0.35, and R^2^_Word-of-Mouth_ = 0.47) indicate a moderately predictive model.

## Figures and Tables

**Figure 1 behavsci-10-00127-f001:**
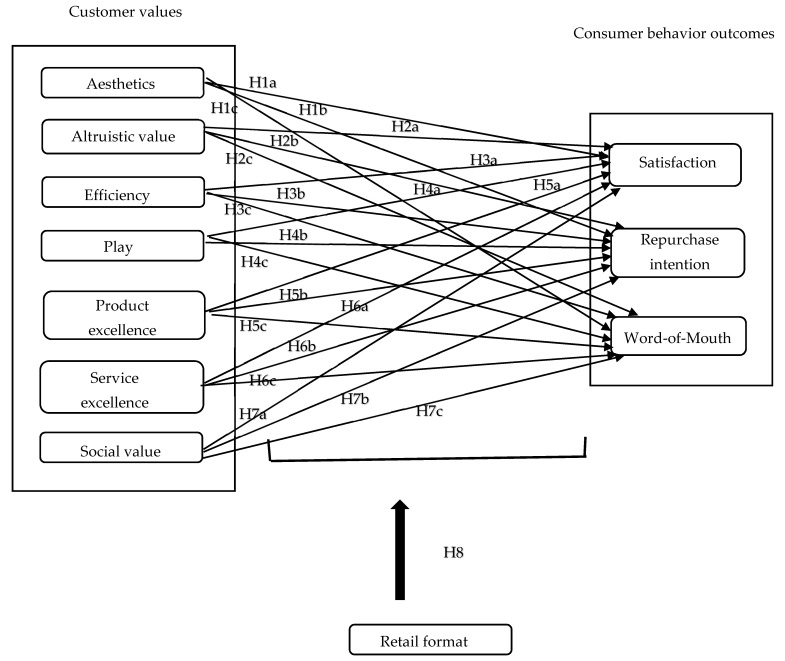
Conceptual model (based on [7]).

**Figure 2 behavsci-10-00127-f002:**
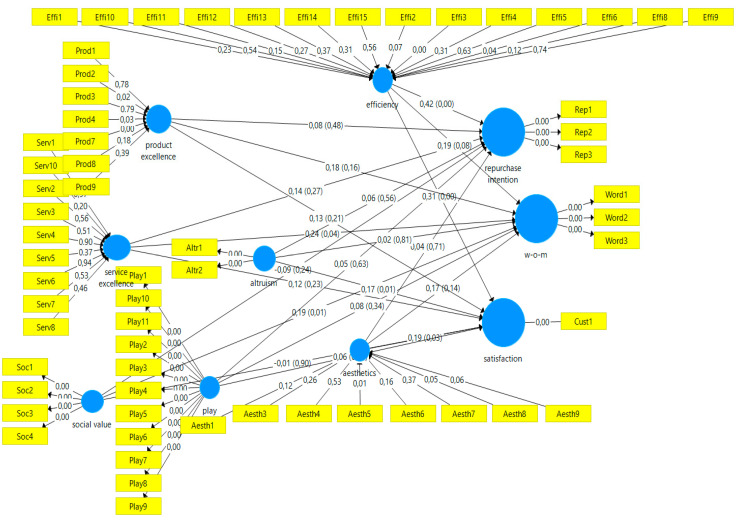
Final model.

**Table 1 behavsci-10-00127-t001:** Overview of customer value types and consumer behavior outcomes, indicator abbreviations, and all scale references.

Constructs	References
Aesthetics (Aesth1-Aesth9)	[7,11,52,53]
Altruistic value (Altr1-Altr2)	[24]
Efficiency (Effi1-Effi15)	[7,11,37,53,54,55,56]
Play (Play1-Play11)	[55]
Product excellence (Prod1-Prod9)	[7,57]
Service excellence (Serv1-Serv10)	[57]
Social value (Soc1-Soc4)	[22]
Satisfaction (Cust1)	[58]
Repurchase intention (Rep1-Rep3)	[54,59]
Word-of-Mouth (Word1-Word3)	[54,59]

**Table 2 behavsci-10-00127-t002:** Classification of the respondents.

Variable	Category	Total Respondents (N = 126)	Percentage
Age	20–30	23	18%
	31–40	39	31%
	41–50	23	18%
	51–60	23	18%
	61–70	10	8%
	70+	8	6%
Gender	man	39	31%
	woman	87	69%
Supermarket	Albert Heijn (=non-discounter)	62	50%
	Plus (=soft discounter)	31	25%
	Lidl (=hard discounter)	33	26%

The systematic evaluation of the results is twofold and consists of the evaluation of the reflective and formative measurement models and the evaluation of the structural model.

**Table 3 behavsci-10-00127-t003:** Cronbach’s alpha, composite reliability, and average variance extracted (AVE).

Constructs	Cronbach’s Alpha	Composite Reliability (CR)	Average Variance Extracted (AVE)
Altruistic value	0.83	0.92	0.85
Play	0.92	0.93	0.56
Social value	0.92	0.94	0.81

**Table 4 behavsci-10-00127-t004:** Indicators’ cross-loadings.

Indicators	Altruistic Value	Play	Social Value
Altr1	0.95	0.32	0.24
Altr2	0.90	0.35	0.10
Play1	0.31	0.81	0.27
Play10	0.28	0.69	0.33
Play11	0.29	0.80	0.28
Play2	0.18	0.72	0.28
Play3	0.33	0.83	0.39
Play4	0.24	0.78	0.15
Play5	0.19	0.69	0.32
Play6	0.23	0.72	0.36
Play7	0.31	0.69	0.34
Play8	0.33	0.77	0.33
Play9	0.29	0.68	0.44
Soc1	0.28	0.44	0.78
Soc2	0.11	0.28	0.94
Soc3	0.17	0.33	0.95
Soc4	0.12	0.37	0.92

**Table 5 behavsci-10-00127-t005:** Fornell-Larcker criterium.

Constructs	Altruistic Value	Play	Social Value
Altruistic value	0.92		
Play	0.36	0.75	
Social value	0.19	0.4	0.89

**Table 6 behavsci-10-00127-t006:** f^2^ effect sizes.

Constructs	Repurchase Intention	Satisfaction	Word-of-Mouth
Aesthetics	0	0.04	0.03
Altruistic value	0	0.04	0
Efficiency	0.18	0.12	0.05
Play	0	0.01	0.01
Product excellence	0.01	0.02	0.04
Service excellence	0.02	0.02	0.08
Social value	0.01	0	0.05

**Table 7 behavsci-10-00127-t007:** Path coefficients.

Constructs	Repurchase Intention	Satisfaction	Word-of-Mouth
Aesthetics	0.04	0.19	0.17
Altruistic value	0.06	0.17	0.02
Efficiency	0.42	0.31	0.20
Play	0.05	0.06	0.08
Product excellence	0.08	0.13	0.18
Service excellence	0.14	0.12	0.24
Social value	−0.09	−0.01	0.19

**Table 8 behavsci-10-00127-t008:** Bootstrapping results of total effects.

Path Coefficients	Original Sample (O)	Sample Mean (M)	Standard Deviation (STDEV)	T Statistics (|O/STDEV|)	*p* Values
aesthetics -> repurchase intention	0.06	0.07	0.19	0.50	0.62
aesthetics -> satisfaction	0.21	0.22	0.09	2.22	**0.03**
aesthetics -> w-o-m	0.20	0.21	0.12	1.59	0.11
altruism -> repurchase intention	0.10	0.08	0.11	0.92	0.36
altruism -> satisfaction	0.18	0.15	0.07	2.35	**0.02**
altruism -> w-o-m	0.07	0.08	0.12	0.59	0.55
efficiency -> repurchase intention	0.49	0.50	0.12	4.08	**0**
efficiency -> satisfaction	0.31	0.31	0.10	2.99	**0**
efficiency -> w-o-m	0.22	0.23	0.11	1.98	**0.05**
play -> repurchase intention	−0.04	−0.02	0.10	0.34	0.73
play -> satisfaction	0.05	0.06	0.09	0.57	0.57
play -> w-o-m	0.02	0.02	0.09	0.26	0.79
product excellence -> repurchase intention	−0.01	0.03	0.13	0.06	0.95
product excellence -> satisfaction	0.11	0.13	0.11	1.09	0.28
product excellence -> w-o-m	0.11	0.14	0.13	0.89	0.37
service excellence -> repurchase intention	0.12	0.14	0.13	0.92	0.36
service excellence -> satisfaction	0.10	0.14	0.11	0.97	0.33
service excellence -> w-o-m	0.29	0.26	0.12	2.38	**0.02**
social value -> repurchase intention	−0.08	−0.10	0.08	1.01	0.36
social value -> satisfaction	−0.03	−0.05	0.09	0.34	0.73
social value -> w-o-m	0.19	0.18	0.09	2.01	**0.05**

**Table 9 behavsci-10-00127-t009:** Overview of hypotheses results.

Hypothesis	Relationship	Effect	Result
H1a	aesthetics -> satisfaction	positive	accepted
H1b	aesthetics -> repurchase intention	positive	rejected
H1c	aesthetics -> w-o-m	positive	rejected
H2a	altruism -> satisfaction	positive	accepted
H2b	altruism -> repurchase intention	positive	rejected
H2c	altruism -> w-o-m	positive	rejected
H3a	efficiency -> satisfaction	positive	accepted
H3b	efficiency -> repurchase intention	positive	accepted
H3c	efficiency -> w-o-m	positive	accepted
H4a	play -> satisfaction	positive	rejected
H4b	play -> repurchase intention	negative	rejected
H4c	play -> w-o-m	positive	rejected
H5a	product excellence -> satisfaction	positive	rejected
H5b	product excellence -> repurchase intention	negative	rejected
H5c	product excellence -> w-o-m	positive	rejected
H6a	service excellence -> satisfaction	positive	rejected
H6b	service excellence -> repurchase intention	positive	rejected
H6c	service excellence -> w-o-m	positive	accepted
H7a	social value -> satisfaction	negative	rejected
H7b	social value -> repurchase intention	negative	rejected
H7c	social value -> w-o-m	positive	accepted
H8	retail format moderates the relation between customer value types and consumer behavior outcomes	ND and SD > HD	rejected

**Table 10 behavsci-10-00127-t010:** Path coefficient results in different retail formats.

	Satisfaction				Repurchase Intention				Word-of-Mouth			
Constructs	Total	ND	SD	HD	Total	ND	SD	HD	Total	ND	SD	HD
Aesthetics	0.21 *	0.2	0.29	−0.16	0.17	0.4	0.07	−0.33	0.17	0.26	0.51	0.28
Altruistic value	0.23 *	0.04	−0.03	0.25	0.02	0.1	−0.03	0.25	0.02	0.21	−0.18	0.12
Efficiency	0.22 ***	0.31	0.49	0.31	0.45 ***	0.49	0.67	0.49	0.22 *	0.22	0.22	0.22
Play	0.06	−0.02	0.04	−0.02	0.08	−0.18	−0.1	0.3	0.08	−0.08	0.07	−0.04
Product excellence	0.13	−0.06	0.21	0.01	0.08	0.05	−0.07	0.3	0.18	0.09	−0.19	0.6
Service excellence	0.12	0.37	−0.08	0.45	0.14	0.05	0.19	−0.01	0.31 *	0.15	0.3	0.13
Social value	−0.01	−0.01	0.07	0.03	−0.1	−0.06	−0.03	−0.09	0.20 *	0.19	0.29	0.27

Note: * *p* < 0.05; ** *p* < 0.01, *** *p* < 0.001; ND = non-discounter, SD = soft discounter, HD = hard discounter.

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
