# Peer review of "Customer Value Types Predicting Consumer Behavior at Dutch Grocery Retailers"

_behavsci, 2020, doi:10.3390/bs10080127_

Round 1
Reviewer 1 Report
The paper is interesting and novel, relevant for of practitioners, policy-makers and scholars, and it is my pleasure to review it.
The paper approaches an interesting topic regarding the customer value predicting consumer behaviour, based on a research on grocery retailers in The Netherlands. Methodology and approaches are interesting, systematic and comprehensive.
In terms of content, scientific research and relevance, the paper is clear and well-constructed, it manages to solve most of the proposed objectives.
However, I would have some considerations and suggestions for improving the quality of the article.
Although there is a part of the introductory aspects and an area of literature review, we consider that these sections should be clearly separated, by declaring intentions (the content) and creating separate subsections 1. Introduction, and, then 2. Literature review.
Likewise, the paper, apparently, does not have a conclusions section, although some of them can be found in the final section - Discussions. Similar, a clear demarcation 4. Discussion then 5. Conclusions is necessary.
Literature review is relatively short and unsystematized on topics, directions, critics and uncovered areas to justify the motivation of this research.
Also, some improvement in style could be done in order to make this section easier to read and understand. E.g,
Row 51-52 Research by [19] showed that satisfaction is higher for 51 fill-in grocery shopping (i.e. when just a few items are needed) than
It shoud be written
Research by Nilsson et al (2017) [19] showed that satisfaction is higher for fill-in grocery shopping (i.e. when just a few items are needed) than ...
... and a few others, in similar way. The type of citation should not affect the fluency and logic of the discourse.
Other formal aspects
The tables must be written more ”compact”, in the present form some are difficult to follow and disturb the fluency of the paper. Also, it should be edited according to mdpi instructions.
Some tables and figures have no title, or it is incorrectly positioned.
The final references are not edited according to mdpi style.
In fact, the entire paper must be revised in formal terms, writing and layout, fonts, etc.
Thank you for the opportunity to review this article and good luck!
Reviewer 2 Report
This is an interesting topic to explore, the research has its merits since the topic is quite new.
It was a pleasure to read this paper, which was informative, well structured, easy to read and understand.
The only thing that I would comment upon is the necessity to be clearer on how the author selected the customers dimensions to include in the model. Further theoretical support would benefit the article.
A conclusion section is missing in the paper. Discussion is important but conclusions are mandatory. Otherwise, what is the purpose of the study?
Good job.
Reviewer 3 Report
This review primary concern is to give an overview of the paper entitled “Customer value types predicting consumer behaviour at Dutch grocery retailers”. In my view, the disadvantages of the current version of the paper far outweigh the advantages, but I am going to do my best to give some pointers to the authors as follows:
- The paper structure is questionable because it does not distinguish the introduction and the review of the literature sections. Besides, it does not distinguish the discussion and conclusion sections either. For these reasons, let me suggest the authors separate the review of the literature from the introduction. Similarly, let me recommend the authors create a conclusion section. A sensible structure moves the writing along logically and gives the chance to include the relevant contents of any scientific paper.
- In the introduction, the authors miss the opportunity of finding ground for their research objectives. Similarly, as the paper does not pin down any research gap, it seems highly recommendable the authors made an effort to identify a specific need of research. By the way, objectives and research gap should be closely related. Please, spot the research gap and set out the research objectives with coherence.
- In the review of the literature, the authors should give careful thought to the following matters:
- Needless to say, that value is a polysemic term. Nevertheless, the authors seem to neglect it by using the different meanings in the same paper and this is the principal cause of the failure. I guess what they want to bring into focus is customer value from an experiential marketing approach, but they talk about cultural values, personal values and principals, perceived values and customer value as if all of this were interchangeable. Not only this dispersion dilutes the authors’ point, but it also turns to be a contradiction in terms and is totally misguided. The answer to this shortcoming lays in thorough reviewing the literature on experiential marketing and being more focused on experiential customer value. Please, deal with one meaning of value, one theoretical framework and go ahead.
- It is also thoughtless of the authors not to find ground for putting forward hypotheses. It goes without saying that these prospective hypotheses should be well-formulated and empirically contrastable. Besides, these potential hypotheses should be inspired in figure 1 model of relationships.
- Concerning figure 1, it is disgraceful to illustrate such an ambiguous model. To be more specific, it is not clear the specific links between every customer value and every consumer outcomes variables. Please, establish the specific connection between both sides of these variables. Also, I wonder what kind of relationship the authors are presuming between the retail format and the above-mentioned variables. Are the authors meaning the existence of a moderating effect or are they assuming a mediating effect? Please, be precise and rigorous.
- This figure should be the core guide of the review of the literature. In other words, it should be the road map of the authors’ writing and address the hypotheses formulation.
- Many shortcomings are stemming from the materials and methods section as follows:
- The sample is small.
- The authors forget to describe the scales of satisfaction, word of mouth and repurchase intention.
- The survey context remains in the dark insofar as the authors do not disclose information about when and who the survey taker performed their job.
- It is not sufficient to classify grocery retailers. I would rather the authors pointed out the criteria they embraced to select the groceries.
- I have no hesitation in recommending that the authors state if their sampling procedure was non-probabilistic and precise what specific type of non-probabilistic procedure they employed.
- It lacks a table in which the authors layout information about the sample profile.
- The results section muddles through the obtained evidence as follows:
- For table 1, what is the meaning of “Altr1, 2, Play” etc? I presume it is dealing with the values items scale but who knows.
- I guess the authors carried out an exploratory factor analysis before they performed a confirmatory factor analysis…did you?
- Table 3 is entitled “indicators’ cross-loadings” but it is not clear what data the authors have laid out in. is a sort of correlation?
- The authors talk about a structural model but they say something about a path coefficients…Please, be consistent.
- Collinearity testing is repeated many times and it should be tackled just once.
- As far as a Structural Equation Modelling is concerned, the authors should carry out a multi-groups analysis rather than perform Anovas’ tests.
- If it were not for the fact that the paper is rather misguided, the discussion section would seem insightful. Nonetheless, the paper lacks a thorough theoretical framework, rigorous methodology and solid evidence. Furthermore, I would like to give some pointers as follows:
- Create a new section for conclusions.
- This conclusion section should preface with a summary, carry on with the paper contribution and append future lines of research. Please, highlight the main paper scientific contribution and do not confuse limitations for future lines of research.
- It would be interesting to provide some practical implications.
- The list of bibliographical references should encompass more recent scientific works and be more focused on experiential value items.
I hope these comments can be of help in improving the paper and encourage the authors to move forward
Round 2
Reviewer 3 Report
This is the second review of the paper entitled “customer value types predicting consumer behaviour at Dutch grocery retailers” and I can state that the paper has improved substantially. To be specific, the authors have separated the introduction and review of the literature sections. Similarly, they have created a new section called conclusions. Secondly, they set out the objectives clearly and found ground for them. Thirdly, they bring up the topic of customer value by considering the experiential literature and, in turn, they avoid interchanging and confusing the concept of value. Fourthly, they describe much better the used measuring instruments by indicating the scale types, their ranges and bibliographical sources. Fifthly, they provide more information about the survey context. Sixthly, they classify the groceries with a theoretical foundation. Seventhly, it is definitively clear that what the authors have performed is a structural equation model with latent variables and not a path model with indicators. Eighthly, there are specific practical implications and the authors acknowledge some limitations. Finally, the authors have included new and relevant bibliographical references.
Nevertheless, there are remaining concerns about a few but important issues such as follows:
- Although the authors acknowledge the appropriateness and importance of putting forward hypotheses, they do it wrongly in that they do neither formulate it correctly nor support it sufficiently. To be specific, let me suggest that the authors address this problem through the following measures:
- Please, formulate the hypotheses (1) completely so that each hypothesis is one sentence. There seem to be seven hypotheses, but it is rather confusing.
- Please, support each hypothesis (1) with particular contents and specific paragraphs. In other words, do not leave the hypotheses (i) all together at the end but rather put forward each of them separately or, alternatively, a set of similar hypotheses together but directly linked to specific supporting contents. I am afraid that, in the current form, it is rather difficult to see how supported every hypothesis is.
- Concerning hypothesis 2, one might question it is well supported. Please, consider that this hypothesis state the existence of a moderating effect of the retail format and the relationship between customer values and consumer behaviour. Nonetheless, nothing is told about how the moderating variable (retail format type) correlates with the dependent variables (consumer behaviour outcomes). The authors ramble about the market share and their marketing mix performances but they do not bring into focus how the moderating variable alters the relationships between independent and dependent variables. What is more, nothing is said about how the moderating variable changes the direction and effect of the independent variables on the dependent variables. Hence, even though hypothesis 2 is well formulated and empirically contrastable, it is not supported at all.
- Although the authors have inserted a figure as regards the conceptual model, this figure is still confusing in that there is not a direct link between the variables. The answer to this problem lays in linking every customer values box (7) to every customer behaviour outcomes box (3). If I have understood it correctly, there must be 21 causal arrows. Please, let us see the specific and structural relationships between the variables and each relationship represents a hypothesis.
- Finally, it is disgraceful to put forward hypotheses and forget to contrast them. Please, describe the hypotheses output by laying it out in a table. In addition to building a table that summarises the hypotheses output, let me suggest that the authors discuss their obtained results by referring to the hypotheses empirical contrast. This contrast exercise might be performed either in the analysis of the results section or discussion section. Are the hypotheses verified or rejected?
- The sample size should be considered a limitation. Please, acknowledge it in the conclusion section.
- There is a mismatch between estimating a structural model and performing an Anova Test. Three multigroups might be carried out instead.
I hope these comments can be of help in improving the paper and encourage the authors to move forward.
Author Response
Dear Reviewer 3, please see the attachment.

Round 3
Reviewer 3 Report
This is the third review of the paper entitled “Customer value types predicting consumer behaviour at Dutch grocery retailers” and while the authors have followed some of my previous indications they refused to adopt others. To be specific, I am glad to acknowledge that Figure 1 has been revised successfully and now the authors are paying attention to contrasting every hypothesis in the analysis of the results section. Nevertheless, there remain some drawbacks as follows:
- Although the authors have formulated the hypotheses separately and, hence, it is much clearer, they fail to express causality in that the term “relate” seems to express a descriptive relationship instead of a causal relationship. Please, replace the term “relate” by the term either “determine” or “influence” or “trigger” or any other causal meaning term. For instance, “H7c: social value influences worth of mouth”.
- All the hypotheses should be supported specifically in this research work. Therefore, let me suggest that the authors make an effort to find ground for the hypotheses in their current paper. If the authors argued that all their hypotheses are already supported by another paper, one might question how worth publishing the current paper is insofar as it would be no other thing than a replication.
- Hypothesis 8 should be reformulated so that it can be empirically contrastable. In my view, hypotheses 8 should be transformed into 21 different hypotheses. Needless to say, each hypothesis should be theoretically and specifically supported. Another option could be to delete this hypothesis, its theoretical contents and empirical evidence. After all, the hypothesis resulted in rejection and I guess it has been due to the fact it was not sufficiently supported.
I hope these comments can be of help in improving the paper and encourage the authors to move forward.
